# Characterizing and Predicting Outcomes in Critically Ill Patients Receiving Low or High Protein Doses with Moderate Energy Support: A Retrospective Study

**DOI:** 10.3390/nu16193258

**Published:** 2024-09-26

**Authors:** Orit Raphaeli, Pierre Singer, Eyal Robinson, Liran Statlender, Ilya Kagan

**Affiliations:** 1Industrial Engineering and Management, Ariel University, Ariel 40700, Israel; 2Data Science and Artificial Intelligence Research Center, Ariel University, Ariel 40700, Israel; 3Institute for Nutrition Research, Felsenstein Medical Research Centre, Petah Tikva 4941492, Israel; pierre.singer@gmail.com; 4Intensive Care Unit, Beilinson Hospital, Rabin Medical Center, Petah Tikva 4941492, Israel; eyalrob@clalit.org.il (E.R.); ilyak@clalit.org.il (I.K.)

**Keywords:** protein, energy, critical care, enteral feeding

## Abstract

Background: Finding the best energy and protein dose and timing for critically ill patients remains challenging. Distinct populations may react differently to protein load. This study aimed to characterize and predict outcomes of critically ill patients who received moderate energy and high or low protein doses during their stay in the intensive care unit (ICU). Methods: The cohort included 646 adult patients (70% men and 30% women) hospitalized in Beilinson Hospital ICU (Petah Tikva, Israel) for over 5 days between 2011 and 2018. Patients received 10–20 kcal/kg/day and were classified into two groups: low (LP) and high (HP) protein support (≤1 g/kg/day vs. >1 g/kg/day), the LP group comprising 531 patients (82%) and the HP group 115 patients (18%). Multiple logistic regression was used to describe associations between patients’ characteristics and 90-day survival in the LP and HP groups. Results: Among LP, increased age, APACHE II, and receiving supplemental parenteral nutrition (SPN) were associated with decreased survival (OR = 0.986, 95% CI [0.973, 0.999]; OR = 0.915, 95% CI [0.886, 0.944], OR = 0.579, 95% CI [0.366, 0.917]). Trauma admission was associated with increased survival (OR = 1.826, 95% CI [1.001, 3.329]). Among HP, increased age was associated with decreased survival (OR = 0.956, 95% CI [0.924, 0.998]). Higher BMI was associated with improved survival (OR = 1.137, 95% CI [1.028, 1.258]). Likewise, in the HP group, the BMI of elderly survivors was higher compared to non-survivors (27.1 ± 6.2 vs. 24.7 ± 4.8, t (113) = 2.3, *p* < 0.05). Conclusions: Our results show that in patients with moderate energy support and low protein administration, survivors were younger, with a lower APACHE II score, mainly suffering from trauma and without renal failure. In the patients receiving high protein support, younger patients with a high BMI not suffering from sepsis were more likely to survive. We suggest confirming these findings with prospective RCTs.

## 1. Introduction

Finding the optimal energy and protein dose, initiation timing, and delivery rate remains challenging [1]. In daily practice, the amount of protein administered is lower than the loss evaluated by urinary nitrogen excretion [2]. Numerous observational studies [3,4,5,6] suggested that increased protein administration was associated with improved survival. In sarcopenic ICU patients, administering more than 1.2 g/kg/day was associated with improved outcomes [7]. The effects on patient outcomes are complex and may vary depending on specific patient factors: In individuals with acute renal injury, more protein supply may not increase mortality but may not generally enhance clinical results [8]. In the ICU, additional protein may have an impact [9,10]. However, these findings were not confirmed by randomized controlled studies [11,12], and meta-analyses found no advantage in the administration of larger doses of protein [13,14]. In addition, the dose and timing of the protein administration, together with the energy administration, are very different from one study to the other [1], making it difficult to come to a uniform conclusion regarding dosing and timing.

Energy and protein administration are related. Some studies have tried to find the optimal dose and timing of protein and energy administration. Using a large international database, Hartl et al. concluded that providing standard protein intake support during the late acute phase may improve outcomes compared to an exclusively low-protein diet. In unselected critically ill patients, clinical outcomes may not be improved by high protein support during the acute phase [15]. There is increasing evidence that a high energy load in the early period of the acute phase may be deleterious [10,16] and that administering moderate energy support may improve outcomes in the ICU [5]. In the study by Lin et al. [17], 2191 patients were included for analysis. A distinct triple-group trajectory of protein support was identified, with 919 patients categorized into the low-level protein support group, 1146 in the medium-level group, and 126 in the high-level group. The mean daily protein supports from the low- to high-level protein support groups during the first week of enrolment were 0.38 ± 0.14, 0.8 ± 0.18, and 1.68 ± 0.39 g/kg/d, respectively. Low-level or high-level protein support in the early phase of critical illness was associated with increased 28-day mortality compared to medium-level protein support. However, when adjusted for energy support, low-support level protein support in the early phase was no longer associated with increased 28-day mortality. In another study, including 1172 patients, Matejovic et al. found that compared to lower daily calory or protein support, moderate support was associated with a higher probability of successful weaning (for calories: maximum HR 4.59 [95% CI: 1.5;14.09] on day 12; for protein: maximum HR 2.60 [1.09;6.23] on day 12), and with a lower hazard of death (for calories only: minimum HR 0.15, [0.05;0.39] on day 19). There was no evidence that high energy or protein support was associated with further outcome improvements. In patients staying in the ICU for ≥5 days, early moderate daily energy and protein intake support were associated with improved clinical outcomes [18]. Different populations suffering from different etiologies (trauma, sepsis, COPD, etc.) may react differently to changed protein loads [19,20]. This study aimed to characterize and predict the outcomes of critically ill patients who received moderate energy and high or low protein doses during ICU stay.

## 2. Materials and Methods

### 2.1. Ethical Approval

This study was approved by the ethics committee of Rabin Medical Center (RMC 0392-14).

### 2.2. Study Design and Participants

This observational retrospective cohort study included all patients aged 18 and above hospitalized in a 16-bed general mixed medical–surgical adult ICU at Beilinson Hospital’s (Petah Tikva, Israel) from 2011 to 2018 for more than five days. Exclusion criteria included patients receiving low (less than 10 kcal/kg/day) or high (more than 20 kcal/kg/day) caloric support. Data were collected from the Beilinson ICU computerized database (Metavision, Israel). A total of 1257 ICU patients had a longer stay (LOS) than five days. Of them, 646 patients received moderate caloric support and were included in our final analysis. Patients were divided into two cohorts based on the average daily protein dose. One was a cohort of 531 patients who received a low protein dose (LP cohort) with an average daily protein of equal to or less than 1 g/kg/day, and the other was a cohort of 115 patients who received a high protein dose (HP cohort) with an average daily protein more than 1 g/kg/day) (Figure 1).

### 2.3. Patient Demographic and Clinical Characteristics

Patient demographics included age, gender, and BMI. Admission data included admission type (medical, surgical, transplantation, and trauma), comorbidities (cardiac, metabolic, neurologic, oncology, pulmonary, renal, and sepsis), and APACHE II score. Clinical nutrition therapy included all enteral (EN) and parenteral (PN) orders administrated during the ICU stay. Clinical outcomes included 90-day survival.

EN and PN orders were summed into daily amounts of kcal and protein. Average daily energy support during the ICU stay over five days was calculated and categorized into three levels: low (<10 kcal/kg/day), moderate (10–20 kcal/kg/day), and high (>20 kcal/kg/day). Average daily protein support over five days was categorized into two levels: high (>1 g/kg/day) and low (≤1 g/kg/day). This limit was taken arbitrarily to reflect the daily practice and follow the medium-level administration described by Lin et al. [9]. There is no consensus regarding the definition of low or high protein administration in the literature.

### 2.4. Statistical Analysis

The baseline characteristics of the HP and LP cohorts were compared using an unpaired *t*-test and the Wilcoxon sum-rank test for normal and non-normal continuous variables. The chi-squared test was used for categorical variables. Statistical significance was considered as a *p*-value < 0.05 for a two-sided test.

In each protein dose cohort, single-variable differences between 90-day survivors and non-survivors were compared using an unpaired *t*-test and the Wilcoxon sum-rank test for normal and non-normal continuous variables. The chi-squared test was used for categorical variables. Multivariate analyses were performed using multiple logistic regression to derive the best-fitting model to describe the relationship between 90-day survival and patient characteristics in both LP and HP cohorts. The odds ratio (OR) was then calculated with a confidence interval of 95%. All statistical analyses were performed using IBM SPSS statistics, version 29.0.0.

## 3. Results

### 3.1. Baseline Characteristics of LP and HP Cohorts

Table 1 presents the baseline characteristics of the study population. The overall cohort included mostly males (70%) with an average age of 60.2 (±17.2), and the leading age group was 40–65 (41%). The average BMI was 28.6 (±6.6) kg/m^2^, with the primary category being overweight (39.6%). The main admission categories include surgical (44.7%) and trauma (34.8%). Most patients received EN (75.4%). Some received EN with supplemental parenteral nutrition (SPN) (22.9%). Few patients received PN (1.7%). The average protein support in the first five days was 0.78 g/kg (±0.39). The main comorbidities include cardiac disease (53.4%), metabolic disorders (51.9%), and kidney disease (39.9%). The 90-day survival rate was 58%. Except for a significantly higher proportion of pulmonary patients in the HP cohort compared to the LP cohort, there were no significant differences in other variables between the cohorts.

### 3.2. Comparing Survivors and Non-Survivors in LP and HP Cohorts

The differences in single variables between survivors and non-survivors in both LP and HP cohorts are shown in Table 2. In the LP cohort, survivors had a significantly lower mean age and APACHE II compared to non-survivors (57.0 ± 18.6 vs. 64.3 ± 14.3, and 21.5 ± 6.2 vs. 25.9 ± 6.5, *p* < 0.001 respectively). Moreover, there was a significantly higher rate of patients with the trauma admission type among survivors compared to non-survivors (38.4% vs. 25.4%, chi-square (3) = 12.4, *p* < 0.005). In terms of comorbidities, there were significantly lower rates of cardiac and renal patients among survivors compared to non-survivors (45.3% vs. 64.3%; chi-square (1) = 18.8, *p* < 0.001; 33.6%, vs. 49.1%, chi-square (1) = 13.0, *p* < 0.001, respectively). No significant differences were observed in other variables. In the HP cohort, survivors had a significantly higher mean BMI and lower age compared to non-survivors (27.2 ± 6.2 vs. 24.7 ± 4.9, *p* < 0.05; and 57.6 ± 18.2 vs. 66.5 ± 12.8, *p* < 0.005, respectively). Moreover, in terms of comorbidities, survivors had a significantly higher frequency of neurological comorbidity (38.2% vs. 21.3%, chi-square (1) = 3.7, *p* < 0.05) and a lower rate of sepsis (2.9% vs. 12.8%, chi-square (1) = 4.1, *p* < 0.005) compared to non-survivors. No significant differences were observed in other variables.

Multivariate logistic regression was performed to ascertain the relationship between patient characteristics and the patient’s likelihood of survival in both protein dose cohorts. In the LP cohort, the model showed that age, APACHE II, SPN route, and trauma admission type were independent predictors of survival (Table 3). These results are similar to the relationships among variables and 90-day survival found with the single-variable analysis. The cardiac and renal comorbidities, which were significant in the single-variable analysis, did not play a significant role in the regression analysis. An increase in age and APACHE II score were associated with a decrease in the likelihood of survival (OR = 0.986, 95% CI [0.973, 0.999]; OR = 0.915, 95% CI [0.886, 0.944]). Patients receiving SPN were less likely to be survivors than EN patients (OR = 0.579, 95% CI [0.366, 0.917]). On the other hand, trauma patients were twice as likely to survive while receiving low protein support (OR = 1.826, 95% CI [1.001, 3.329]). The logistic regression model was statistically significant, chi-square (16) = 90.05, *p* < 0.001. The model explained 21% (Nagelkerke R^2^) of the variance in low protein support and correctly classified 67.6% of cases.

In the HP cohort, the model showed that age and BMI were independent predictors of survival (Table 3). These results are similar to the relationships among variables and 90-day survival found with single variable analysis. Neurologic comorbidity and trauma admission type, which were significant in the single-variable analysis, did not play a significant role in the regression analysis. An increase in age was associated with a decrease in the likelihood of survival (OR = 0.956, 95% CI [0.924, 0.998]). In contrast, an increase in BMI was associated with an increase in the likelihood of survival (OR = 1.137, 95% CI [1.028, 1.258]). The logistic regression model was statistically significant, chi-square (15) = 34.11, *p* < 0.005. The model explained 34.1% (Nagelkerke R^2^) of the variance in high protein support and correctly classified 76.5% of cases.

### 3.3. Sensitivity Analysis

#### 3.3.1. Renal Patients

Among the 258 renal patients, 45 received high protein, and 213 received low protein. No significant differences were observed in any of the variables between survivors and non-survivors in both the HP and LP cohorts (*p* > 0.05). Multivariate logistic regression was not statistically significant in both cohorts (chi-square (15) = 15.9, *p* > 0.05, chi-square (15) = 24.99, *p* > 0.05, respectively).

#### 3.3.2. Elderly Patients in the HP Cohort

Following the finding that survivors had a lower mean age than non-survivors in the HP cohort, we examined the differences in the BMI of elderly (age > 65) survivors who received high protein. Results of the *t*-test analysis revealed that the BMI of elderly survivors who received high protein was higher compared to the BMI of non-survivors (27.1 ± 6.2 vs. 24.7 ± 4.8, t (113) = 2.3, *p* < 0.05).

## 4. Discussion

Our results show that in patients with moderate energy support and low protein administration, survivors were younger, with a lower APACHE II score, mainly suffering from trauma, and without renal failure. In the patients receiving high protein support, again, younger patients, not suffering from sepsis, were more likely to survive, as well as patients with a high BMI. Interestingly, an age over 65 with a high BMI was also associated with a better outcome. An elevated BMI seemed to be associated with decreased mortality in patients receiving high protein support, and pulmonary patients’ risk of death was reduced while receiving high protein support. Our approach is based on real-life protein administration to different populations and attempts to individualize populations that may benefit from protein administration beyond the results of large RCTs. Our findings demonstrate that the clusters of patients with younger age, a higher BMI, with trauma, and without sepsis had improved survival when receiving high protein administration.

Age is a well-known prognosis feature in critically ill patients, as demonstrated in the APACHE II and NUTRIC scores [21,22]. Our analysis confirms the importance of age in the prediction of mortality. A scoping analysis in older patients stressed the advantages of higher protein support [23]. However, others found that older people above 76 years old were at risk for higher 28-day mortality [24]. In another study of elderly mechanically ventilated critically ill patients, those who achieved 80% of prescribed EN calories had lower ICU and hospital mortality. Increased EN protein support only lowered hospital mortality [25].

If age is an important and well-recognized prognostic factor, our results suggest that in patients with a higher BMI and maybe more muscle mass, administration of a larger amount of protein may be beneficial. It has been observed that in the low-BMI (<18.5 kg/m^2^) population, there is an association with increased mortality, but an increase in energy and protein delivery in patients with elevated APACHE II and NUTRIC scores was associated with an improvement in survival in a large population studied by Compher et al. [26]. However, our population’s BMI was not low. Even in obese patients suffering from sepsis, a meta-analysis has shown that overweight or obese BMIs reduce adjusted mortality [27]. When an elevated BMI was related to protein support, a recent post hoc analysis of the EFFORT Protein trial did not find significant differences between patients prescribed 2.2 g/kg/day and 1.2. g/kg regardless of the BMI [28]. In our study, however, a protein support of 2.2 g/kg/d was not reached. Bendavid et al. found that in the Nutrition Day ICU, patients are fed regardless of their BMI, leading to overfeeding of the very lean patients up to 30 kcal/kg/day and underfeeding the severely obese patients [29]. An increased BMI may prevent fmuscle loss by attenuating inflammation and protein catabolism [30].

Our cohort includes critically ill patients with different comorbidities and admission etiologies. Our analysis found that patients with pulmonary and cardiac comorbidities could benefit from high protein support. This population is usually not well investigated, and, therefore, our findings may generate some prospective studies in this category. Such a population has been evaluated in a post hoc analysis of the NOURISH study [31] (not an ICU study), where the authors found that a high-protein oral nutritional supplement (20 g protein per serving) containing beta-hydroxy-beta-methyl-butyrate (HMB) reduced mortality compared to a protein-free placebo in a subgroup of older, malnourished patients with COPD. The 30-, 60-, and 90-day mortality risk was approximately 71% lower with high protein supplementation relative to the placebo (1.83%, 2.75%, 2.75% vs. 6.67%, 9.52%, and 10.48%, *p* = 0.0395, 0.0193, 0.0113, resp.), along with a significant improvement in handgrip strength. The authors concluded that their findings support the early administration of high-protein ONS to help preserve muscle mass and improve clinical outcomes in older, malnourished patients after hospital admission [31]. However, this population has not yet been studied separately in large prospective randomized trials and should be the subject of more investigations. Our study also showed that additional protein is not beneficial in patients with renal failure, confirming the recent post hoc analysis of the EFFORT trial by Stoppe et al. [32]. In the subgroup analysis in critically ill patients with AKI, the usual dose (average 0.9 g/kg/d) of protein administration was beneficial in terms of reduced mortality and time to discharge alive from the hospital compared to high-dose protein (average 1.5 g/kg/d). Our study also confirms the advantages of protein administration in trauma patients already observed by others. A higher protein support was generally associated with an improved nitrogen balance, yet many patients had a negative nitrogen balance [33]. Trauma patients who did not achieve their protein targets within a week suffered from more abdominal infection, peritonitis, bowel resection, intestinal fistula, or septic shock [34]. Hartwell et al. also described a decrease in the complication rate when protein goals were reached within four days after admission [35].

Finally, sepsis patients may not be able to utilize higher protein amounts for anabolism, as shown by Weijs et al. [19]. Compared to non-septic patients who improved survival with early higher protein support (<0.8 vs. 0.8–1.0 vs. 1.0–1.2 g/kg/day), septic patients had no beneficial effect from higher protein support at day 4, confirming the anabolic resistance observed in this population. Others found that in sepsis patients, medium (0.8–1.2 g/kg/d) protein support at days 4–7 was associated with lower 6-month mortality (hazard ratio [HR]: 0.646, 95% confidence interval [CI]: 0.418–0.996, *p* = 0.048) compared with high (>1.2 g/kg/d) support [36]. On the other hand, Elke et al., in a secondary analysis of a large nutrition database, described a decrease in mortality and length of ventilation when protein administration was increased by 30 g/d (mean support was 49 g/d corresponding to 0.7 g/kg/day) in critically ill septic patients in the early phase of ICU stay [37]. Our analysis confirms these previous studies. Interestingly, patients requiring SPN had a higher mortality. This population suffers from gastrointestinal intolerance that requires the prescription of SPN. This association between poor outcomes and GIT intolerance has been largely documented [38,39].

### Limitations and Advantages

Our data have been extracted retrospectively from a single-center database and have not been validated in databases from other centers. They may only reflect the studied population; to be generalized, they need to be confirmed and validated in other populations. In intensive care, patients are very different from each other, even with the same diagnosis. In addition, the clinical trajectory may vary, and a multitude of decisions are taken every day for every patient, making general recommendations not accurate enough. Our findings are aimed to generate hypotheses and promote studies that may confirm some of our findings. The advantage of our study is its ability to integrate nutrition dose, timing, and route in a large population, trying to untangle complexity.

Two large prospective randomized studies (EFFORT Protein and PRECISe) [12,40] and a very recent review from Bear et al. [41] have been published recently, reaching the conclusion that high protein intake has no advantage over the ESPEN recommended protein dose of 1.3 g/kg/d. In addition, a previous study comparing 1.3 g/kg/day to a lower dose [42] did not find any significant difference between the groups, but the recruited patients were in shock and received full feeding from day 2, a non-usual nutritional procedure. In addition, many of these patients received TPN. Most of the authors concluded from these studies that there is no advantage to high protein doses. However, Bear et al. [41], in their recent review, pointed out that “Future work is likely to explore the role of protein in specific patient populations such as those with persistent critical illness or obesity”. One of the pitfalls of a large PRCT is the recruitment of large ICU population without discrimination in subpopulations.

Our study focuses exactly on this possibility to find specific patient populations. The cited recent literature based on prospective controlled studies or post hoc analysis failed to find an improvement in outcome using higher doses of protein in all populations and in acute kidney injury or in obesity. Some of our findings are not new, such as the association between better survival and lower age, lower APACHE II, and patients suffering from trauma but not suffering from sepsis. Some of our new findings are as follows: (a) An increased BMI was associated with an increase in survival in the high-protein group. (b) An increased BMI in the elderly, in the high-protein group, was also significantly associated with increased survival. (c) The comorbidity aspect analyzed in our “true-life” study showed that in the low-dose protein group, the patients suffering from cardiac and renal comorbidities had worse outcomes, and in the high-protein regimen, the patients suffering from neurological comorbidities survived better, generating new hypotheses for research.

## 5. Conclusions

Determining the appropriate protein and energy dose for critically ill patients is challenging. Results differ in the different studies. The differences in patient populations, intervention protocols, and study designs may contribute to these disparate results. In the management of critically ill patients, appropriate energy and protein intake is crucial for recovery and prognosis. Compared with large prospective randomized studies comparing various protein doses in critically ill patients, our retrospective approach suggests that patients with several phenotypes can have improved outcomes when receiving moderate energy support and higher protein administration. Some of our results confirm previous studies. A better outcome was observed in younger patients and those with a higher BMI. Interestingly, patients with co-morbidities such as pulmonary or cardiac diseases had better survival when receiving high protein support. Finally, elderly patients with a higher BMI also benefitted from higher protein support. We suggest confirming these findings with prospective randomized studies.

## Figures and Tables

**Figure 1 nutrients-16-03258-f001:**
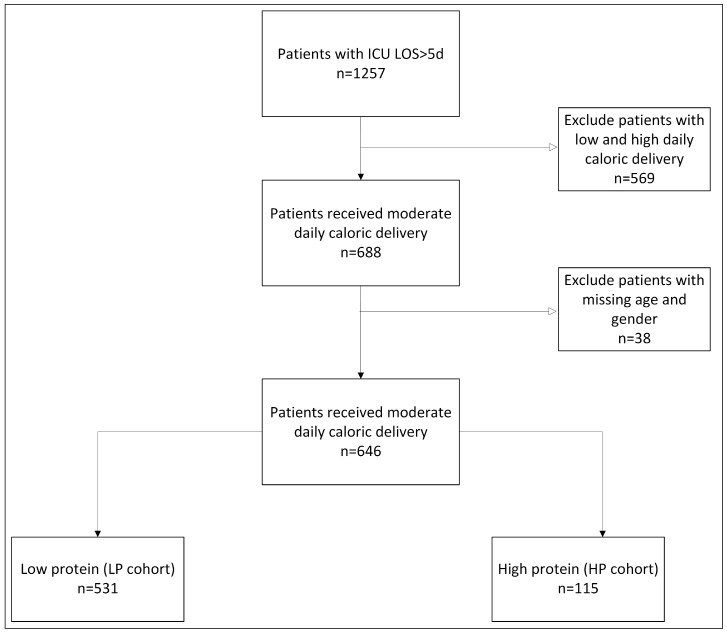
Study cohort.

**Table 1 nutrients-16-03258-t001:** Baseline characteristics of LP and HP cohorts.

Variable ^1^	Overall Cohort (*n* = 646)	LP Cohort (*n* = 531)	HP Cohort (*n* = 115)
Age	60.2 (±17.2)	60.1(±17.3)	60.6 (±16.9)
Gender Male	453 (79.1%)	374 (70.4%)	79 (68.7%)
BMI (kg/m^2^)	28.6 (±6.6)	29.1 (±6.6)	26.2 (±5.8)
Admission Type			
Medical	96 (14.9%)	81 (15.3%)	15 (13%)
Surgical	289 (44.7%)	246 (46.3%)	43 (37.4%)
Transplantation	36 (5.6%)	29 (5.5%)	7 (6.1%)
Trauma	225 (34.8%)	175 (33%)	50 (43.5%)
APACHE II	23.1 (±6.6)	23.4 (±6.7)	21.8 (±5.9)
Nutrition therapy			
EN	487 (75.4%)	404 (76.1%)	83 (72.2%)
EN and SPN	148 (22.9%)	116 (21.8%)	32 (27.8%)
PN	11 (1.7%)	11 (2.1%)	-
Average protein (kg/d)	0.78 (±0.39)	0.68 (±0.15)	1.3 (±0.67)
Cardiac	345 (53.4%)	283 (53.3%)	62 (53.9%)
Metabolic	335 (51.9%)	276 (52.0%)	59 (51.3%)
Neurologic	180 (27.9%)	144 (27.1%)	36 (31.3%)
Oncology	141 (21.8%)	116 (21.8%)	25 (21.7%)
Pulmonary	232 (35.9%)	179 (33.7%)	53 (46.1%) *
Renal	258 (39.9%)	213 (40.1%)	45 (39.1%)
Sepsis	45 (7%)	37 (7%)	8 (7%)
90-day survival	375 (58%)	307 (57.8%)	68 (59.1%)

^1^ Continuous variables are displayed as means and standard deviations. Dichotomous variables are displayed as the number of patients and percentage of the protein group. LP—low protein; HP—high protein; BMI—body mass index; EN—enteral nutrition; SPN—supplemental parenteral nutrition; PN—parenteral nutrition; * *p* < 0.05.

**Table 2 nutrients-16-03258-t002:** Characteristics of 90-day survivors and non-survivors in LP and HP cohorts.

Variable ^1^	LP Cohort (*n* = 531)		HP Cohort (*n* = 115)	
	90-Day Survivors (*n* = 307)	Non- Survivors (*n* = 224)	90-Day Survivors (*n* = 68)	Non- Survivors (*n* = 47)
Age	57.0 (±18.6) *	64.3 (±14.3)	56.6 (±18.2) *	66.5 (±12.8)
Gender Male	214 (69.7%)	160 (71.4%)	46 (67.6%)	33 (70.2%)
BMI (kg/m^2^)	28.9 (±6.1)	29.5 (±7.3)	27.2 (±6.2) *	24.7 (±4.9)
Admission type				
Medical	40 (13%)	41 (18.3%)	8 (11.8%)	7 (14.9%)
Surgical	137 (44.6%)	109 (48.7%)	23 (33.8%)	20 (42.6%)
Transplantation	12 (3.9%)	17 (7.6%)	2 (2.9%)	5 (10.6%)
Trauma	118 (38.4%) *	57 (25.4%)	35 (51.5%) *	15 (31.9%)
APACHE II	21.55 (±6.3) *	25.9 (±6.5)	21.1 (±6.0)	23.0 (±5.6)
Nutrition therapy				
EN	243 (79.2%)	161 (71.9%)	49 (72.1%)	34 (72.3%)
EN+SPN	57 (18.6%)	59 (26.3%)	19 (27.9%)	13 (27.7%)
PN	7 (2.3%)	4 (1.8%)	-	-
Average protein (kg/d)	0.68 (±0.2)	0.67 (±0.2)	1.2 (±0.3)	1.4 (±0.9)
Cardiac	139 (45.3%) *	146 (64.3%)	34 (50%)	28 (59.6%)
Metabolic	150 (48.9%)	126 (56.3%)	32 (47.1%)	27 (57.4%)
Neurologic	92 (30%)	52 (23.2%)	26 (38.2%) *	10 (21.3%)
Oncology	66 (21.5%)	50 (23.2%)	11 (16.2%)	14 (29.8%)
Pulmonary	97 (31.6%)	82 (36.6%)	30 (44.1%)	23 (48.9%)
Renal	103 (33.6%) *	110 (49.1%)	23 (33.8%)	22 (46.8%)
Sepsis	17 (5.5%)	20 (8.9%)	2 (2.9%) *	6 (12.8%)

^1^ Continuous variables are displayed as means and standard deviations. Dichotomous variables are displayed as the number of patients and percentage of the protein group. LP—low protein; HP—high protein; BMI—body mass index; EN—enteral nutrition; SPN—supplemental parenteral nutrition; PN—parenteral nutrition; * *p* < 0.05.

**Table 3 nutrients-16-03258-t003:** Multivariate relationship of 90-day survival and significant baseline predictors in LP and HP cohorts.

Cohort	Variable	Odds Ratio	95% CI	*p*-Value
LP	age	0.986	[0.973, 0.999]	0.04
	APACHE II score	0.915	[0.886, 0.944]	0.001
	EN + SPN route (reference category is EN)	0.579	[0.366, 0.917]	0.02
	Trauma admission type (reference category is medical)	1.826	[1.001, 3.329]	0.05
HP	age	0.956	[0.924, 0.998]	0.008
	BMI	1.137	[1.028, 1.258]	0.013

LP—low protein; HP—high protein; BMI—body mass index; EN—enteral nutrition; SPN—supplemental parenteral nutrition.

## Data Availability

Data are unavailable due to privacy and ethical restrictions.

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
