# Peer review of "Characterizing and Predicting Outcomes in Critically Ill Patients Receiving Low or High Protein Doses with Moderate Energy Support: A Retrospective Study"

_nutrients, 2024, doi:10.3390/nu16193258_

Round 1
Reviewer 1 Report (New Reviewer)
Comments and Suggestions for Authors
Authors presented an observational retrospective cohort study to characterize patients who benefitted from moderate energy and low/high protein delivery during the stay in a large mixed ICU population. The study's findings demonstrated that survivors were younger, had a lower APACHE II score, were primarily traumatized patients, and did not have renal failure when they received moderate energy support and low protein delivery. Younger patients without sepsis and with a high body mass index (BMI) had a higher chance of surviving among the patients getting high protein therapy.
[I suppose ICU is intensive care unit, please explicit it during the first use]
The study is interesting and well written.
The available data indicates that although consuming more protein is advised, its effects on patient outcomes are complex and may vary depending on specific patient factors: In individuals with acute renal injury, more protein supply may not increase mortality but may not generally enhance clinical results (10.1016/j.clnu.2024.07.018; . The role of nutritional supplementation is crucial (10.1016/j.jnha.2024.100256), in ICU it has an impact on prognosis (10.3390/brainsci12091232). The topic is very controversial (10.7717/peerj.17433; 10.1186/s13054-023-04783-1).
Author Response
Thank you very much for taking the time to review this manuscript. Please find the detailed responses in the file attached and the corresponding revisions/corrections highlighted/in track changes in the re-submitted files

Reviewer 2 Report (New Reviewer)
Comments and Suggestions for Authors
The manuscript is a retrospective analysis of a cohort including ICU patients to compare the outcome in high and low protein dose groups.
The title and the introduction seem misleading. The focus of the study is on protein dose in patients with a "moderate" caloric intake. In fact, the energy supply is low (< 20 kcal/kg/d) compared to the recommandations. It might be interesting to study the outcome of patients depending on protein dose in this context. However, the results show prognosis factors in the low and high protein dose groups apart but the effect of the protein dose in unknown.
In addition, the protein dose cut-off was arbitrary (not based on previous studies on protein dose in ICU patients), limiting the value of the results.
The findings of the study are "survivors were younger, with a lower APACHE II score, mainly suffering from trauma and without renal failure." I am not sure that this is new information. Then, the comparision between groups is based on regression analysis in each group and computation of odd-ratios. I do not think it allows for between-groups comparisons.
Finally, a study on the outcome of patients with lower than recommanded caloric intake is debattable or the authors would have to give strong arguments for this management (20 kcal/kg/d would be acceptable but how many patients received 10-15 kcal/kg/d ?). There may be a difference in outcome between 20 and 15 or less kcal/kg/d. I suspect that the lowest caloric intake was associated with the lowest protein dose. In this case, both factors have to be analyzed. Regarding the protein dose, I suggest that the focus should be to compare the outcome between the high and low protein dose, using an appropriate statistical design (propensity score for example).
Comments on the Quality of English LanguageGood quality
Author Response
Thank you very much for taking the time to review this manuscript. Please find the detailed responses in the file attached and the corresponding revisions/corrections highlighted/in track changes in the re-submitted files

Reviewer 3 Report (New Reviewer)
Comments and Suggestions for Authors
1. Shouldn't consideration be given to specific diseases?
2. It is natural that there would be a large difference in baseline protein amount in the age range of 40 to 118 years old, but I would like to know why this age range was set.
3. I would like some description of what the study will ultimately contribute to.
Author Response
Thank you very much for taking the time to review this manuscript. Please find the detailed responses in the file attached and the corresponding revisions/corrections highlighted/in track changes in the re-submitted files

Round 2
Reviewer 2 Report (New Reviewer)
Comments and Suggestions for Authors
Thank you, no further change required
This manuscript is a resubmission of an earlier submission. The following is a list of the peer review reports and author responses from that submission.
Round 1
Reviewer 1 Report
Comments and Suggestions for Authors
This manuscript examined ICU patient’s characteristics receiving moderate energy and low/high protein support in order to establish which patients benefited from them. Overall, although the topic and results are of interest, some improvements should be made:
Abstract section:
- Line 13: “Different populations may react differently to protein load.” You should try to use synonyms to make the reading more attractive. In this case, you could use “distinct populations”, for example.
- Lines 15-16: “Beilinson Hospital ICU”. I would indicate the city and country in parentheses.
- Line 18: change <= for ≤ (and throughout the text)
- Line 19: “Of the 646 ICU patients, 531 (82%) received LP, and 115 (18%) 19 received HP”. This information can be summarized previously, where is lacking.
E.g.: […]The cohort included 646 adult patients (--% men and --% women) hospitalized in Beilinson Hospital ICU (city, country) longer to 5 days, between 2011-2018. Patients received 10-20 kcal/kg/day and were classified in 2 groups: the low (LP) and high (HP) protein support (≤ 1 g/kg /day vs. >1 g/kg /day), comprising the LP group 531 patients (82%) and the HP group 115 patients (18%) […].
- I wouldn’t use the term “intake” (line 18) because it implies some voluntary actions and most of these patients can’t do it. The term “support” is more appropriate. Please change it and revise throughout the text the correct use.
- Lines 25-26: change +/- for ± (and throughout the text)
- You should provide some specific conclusions according to your results in the last sentences. For instance, you referred in your conclusion that younger patients, with higher BMI or with co-morbidities appears to benefit from higher protein support.
2Introduction section:
It must be maturated and rewritten. This section is more likely to the discussion section. It must be a contextualization, un updated background more descriptive (there are so many data), going from the current knowledge to the gaps in the literature, with some hypothesis before the general aims.
Materials and Methods Section:
- Please, consider my previous suggestions for abstract also in this section.
- You must provide information regarding Ethical Committee.
- Description and figure of “Study design and cohort description” could be included in this section instead of in results, since it provides basic information about your sample size and the inclusion/exclusion criteria.
Results sections:
- Please make a deep review of mathematics symbols/expressions. In addition to previously mentioned, in line 104, units of BMI (kg/m2) the number 2 must be superscript. These small details are important and you must expressed correctly in the text.
- All tables and figures must be self-explanatory, which means you have to provide always an abbreviation list including all the abbreviatures used in that table/figure (no matters if you have previously specified in the text).
- Line 119 “p<.001” please add the first 0; this expression is not mathematically correct.
- Figure 3, 4, 5 and 6 should be improved. In figure 3, I would remove parallel lines to the x-axis. The size of the numbers and letters is too small, difficult to read and thus, to understand. Although the most important thing is the data/results, aesthetic is also important. Moreover, you could consider if some of these figures/results could be presented together to reduce the number of figures (just a suggestion; choose the option in which the results are presented clearer).
Discussion section:
- The first paragraph should end in line 199 “large RCTs. // Age is a”, because it must be a summary of your main results and maybe a last sentence with their potential implications. And then, in the following paragraphs, you discuss these main outcomes (and related secondary outcomes) in an ordered way.
- In general terms the discussion is acceptable but you could improve it if you reorder a extend a little bit some parts, looking for some possible explanations of your findings, practical implications, etc.
- Limitations list could be amplified. If you check other related articles you will see how many limitations exist in this kind of research, especially with critical ill patients.
Conslusion section:
Line 265: “Compared with large prospective randomized studies comparing various”. Please, avoid the use of similar words/terms (compared/comparing) in the same sentence or following sentences throughout the text to improve readability.
Comments on the Quality of English LanguageWriting can be improved, especially for doing more attractive the reading process, but it is acceptable.
Reviewer 2 Report
Comments and Suggestions for Authors
Dear Authors,
I congratulate you for your effort in finding the best energy and protein dose administration and timing for critically ill patients. Unfortunately, your study could not clarify this aspect or bring new data; it only confirms the importance of age in the prediction of mortality. It also confirms the results of other similar studies regarding the administration of proteins in critical patients with sepsis, cardiac, pulmonary or renal diseases.
Abstract: -line 20: please define SPN at the first appearance in the abstract and in the main text.
Introduction: - line 31: The introduction is very abrupt, without specifying the population to which the study refers.
The purpose of the study is not well defined; please specify what are the main objectives of the study, as well as the secondary ones.
Materials and Methods: Please specify more clearly how the patients were divided into subgroups, how they were analyzed, and which comorbidities were studied.
The Results chapter could benefit from a structuring of sub-chapters, for a better visualization and follow-up of the data.
Figure 6 is redundant; the results are very well described in the text.
Discussion: In this chapter, it would be better if you highlighted the statistically significant results of your study.
Round 2
Reviewer 1 Report
Comments and Suggestions for Authors
The authors have carried out all the suggested changes. Nothing to add.
Comments on the Quality of English LanguageEnglish quality is aceptable
Reviewer 2 Report
Comments and Suggestions for Authors
Dear Authors,
Thank you for the answers to my comments and for the effort put into conducting such a study. Unfortunately, the results are predictable and already known/proven in larger studies; the news brought by you does not seem important enough to me to merit publication in the Nutrients journal.